# Capturing Structure and Feature Signals in Graph Self-Supervised Learning

## Abstract

This paper analyzes graph self-supervised learning (SSL) methods for node-level prediction tasks. First, we thoroughly evaluate several representative SSL methods on a diverse set of graph datasets. We observe that, contrary to prior literature, two popular generative methods MaskGAE and GraphMAE often fail to outperform well-tuned supervised baselines. At the same time, the contrastive methods BGRL and GRACE on average perform better than generative methods and supervised baselines. We hypothesize that this happens because BGRL and GRACE are able to capture the information about both graph structure and node features, while MaskGAE and GraphMAE concentrate on a single source of information. We support this hypothesis by conducting an analysis on carefully designed synthetic data. Motivated by our observations, we recommend designing SSL objectives that capture both feature and structure information. To verify the effectiveness of this approach, we propose a generative method that reconstructs both graph structure and node features. While being simple, this method outperforms all other considered approaches.

## 1 Introduction

Self-supervised learning (SSL) is a powerful paradigm for learning representations in various domains, including computer vision, natural language processing, and graph-structured data. SSL allows for producing useful representations from unlabeled data in a task-agnostic way that may be especially beneficial when the amount of labeled data is limited. In the context of graphs, SSL leverages the inherent structure of data to learn meaningful node- or graph-level embeddings that can be adapted for various downstream tasks such as node and graph classification or link prediction (Wu et al., 2021; Liu et al., 2022; Xie et al., 2022).

In this work, we start with a thorough evaluation of several representative graph self-supervised learning methods with a focus on node-level prediction. Our analysis is conducted on 10 diverse datasets from various domains, covering both homophilic and heterophilic graphs with homogeneous and heterogeneous node features. We pay special attention to carefully tuning both supervised baselines and the considered graph SSL methods. For this, we employ thorough hyperparameter search that has received insufficient attention in prior works, and utilize architectural enhancements that have proven to be effective for graph learning problems.[1]

Our evaluation reveals a surprising outcome. Under the standard evaluation protocol of linear probing, representative generative methods, namely, MaskGAE and GraphMAE, often fail to outperform well-tuned supervised baselines, while contrastive methods GRACE and BGRL often have better performance than both supervised baseline and generative methods. At the same time, full finetuning allows all the considered methods to surpass the supervised baseline, proving that they are able to capture some useful information.

One would expect that whether a self-supervised learning method has a better or worse performance, especially under linear probing, is related to the amount of useful signal captured during the pretraining phase. Thus, to deepen our understanding of graph SSL methods, we conduct analysis on carefully designed synthetic data that allows for investigating what type of information is captured by a given model. Specifically, we analyze the quality of node features and graph structure reconstruction

---

[1]Our code is available at https://anonymous.4open.science/r/12ab85e0a9c0dbcc.

from the model embeddings. We find that MaskGAE and GraphMAE primarily focus on a single characteristic, either graph structure or node features. Meanwhile, contrastive methods are capable of capturing information from both sources. In particular, we observe that contrastive methods with feature augmentations can capture information about graph structure, while structure augmentations lead to improved node feature reconstruction. We argue that this ability to capture both types of signals is the reason for good performance of the considered contrastive methods.

Based on our observations, we recommend designing future graph self-supervised learning methods so that they capture information about both graph structure and node features. To verify the validity of this approach, we propose a simple generative method that directly follows our recommendation. We name this method GrASP: Graph Attribute and Structure Prediction. Its naive version GrASP$_{naive}$ directly combines MaskGAE and GraphMAE losses, while preserving all their design choices. This naive version already performs better than individual MaskGAE and GraphMAE methods, and nearly matches or even exceeds the performance of the supervised baseline. By further reducing the complexity of certain design choices, we obtain GrASP and show that it outperforms all other considered approaches, while being surprisingly simple.

Overall, our contributions can be summarized as follows:

- We conduct a thorough empirical comparison of graph SSL methods and show that better benchmarking practices can affect the evaluation results.
- We provide an analysis of information captured by different methods and its effect on downstream performance, using both real and synthetic data.
- Based on our insights, we give a very simple yet often overlooked recommendation: graph SSL methods should capture information about both graph structure and node features.
- We propose GrASP, a novel, simple, and effective generative method that explicitly captures both information types, outperforming all other considered approaches.

## 2 BACKGROUND ON GRAPH SELF-SUPERVISED LEARNING

Graph self-supervised learning is a powerful technique that allows one to effectively exploit unlabeled data, which is especially useful when the labeled data is scarce. There are many works dedicated to this topic — see, for example, the following surveys by Wu et al. (2021); Xie et al. (2022); Liu et al. (2022); Zhao et al. (2024b). Arguably, two key approaches in graph SSL are contrastive and generative (Wu et al., 2021).

**Contrastive approaches**  Contrastive methods learn representations by contrasting similar and dissimilar pairs (of nodes or graphs), and many contrastive methods heavily rely on augmentations for generating similar pairs (Veličković et al., 2018; Zhu et al., 2020b; Hassani & Khasahmadi, 2020; Zhu et al., 2021a). Employing appropriate augmentations can be crucial, thus several works aim to improve the quality of augmentations (Zhu et al., 2021b; You et al., 2021), while other works propose to alleviate the need for augmentations at all (Xia et al., 2022). Another challenge in graph contrastive learning is scalability, as classic contrastive methods require a large number of negative samples in order to work well. Thus, another line of research is dedicated to alleviating the need for negative samples (Thakoor et al., 2022; Bielak et al., 2022; Sun et al., 2024).

**Generative approaches**  Generative methods learn to generate or reconstruct data like graph structure and node features. While these methods originated with (variational) graph autoencoders (Kipf & Welling, 2016a), further focus has moved to masked autoencoders, which employ partial input masking in order to make the reconstruction task more challenging (Hou et al., 2022; Li et al., 2023; Hou et al., 2023; Zhao et al., 2024c). Other generative approaches include autoregressive reconstruction (Hu et al., 2020) and diffusion-based methods (Yang et al., 2024; Chen et al., 2025).

**Representative methods**  Let us describe in more detail some specific representative methods of both generative and contrastive approaches that we consider in this work.

GRACE (Zhu et al., 2020b) is a classical graph contrastive learning method. It first generates two correlated views of a graph by applying augmentations and then leverages contrastive loss for node-level representations. If two nodes from two augmented views were obtained from the same node in

the initial graph, they are considered to be a positive pair, and all the other node pairs are treated as negative pairs. For augmentation, GRACE employs edge masking that uniformly at random removes some edges from a graph, and node feature masking that samples a subset of features and replaces them with zeros for all nodes.

BGRL (Thakoor et al., 2022) is an adaptation of standard BYOL (Grill et al., 2020) to the graph domain, and it is another representative contrastive learning method. BGRL effectively alleviates the need for negative examples by leveraging the bootstrap mechanism to obtain useful representations. More specifically, BGRL proposes to learn an online GNN by predicting the outputs of the target GNN, and the target GNN is constructed as the exponential moving average of the online GNN. The online and target GNNs have two different augmented views of the initial graph as inputs. For augmentation, BGRL utilizes edge masking and node feature masking, similarly to GRACE.

MaskGAE (Li et al., 2023) is a representative generative method that utilizes structure reconstruction as an SSL task. MaskGAE selects a mask of edges using either uniform sampling or random-walk-based sampling. The selected edges are removed from the graph. After that, a GNN is applied to the remaining graph in order to obtain the node representations, which are further used to reconstruct masked edges and predict node degrees in the masked graph.

GraphMAE (Hou et al., 2022) is another representative generative method. Unlike MaskGAE which relies on reconstructing structural information, GraphMAE leverages feature reconstruction as an objective. Specifically, it first samples a set of nodes and replaces all features in these nodes with a learnable mask. Then, a GNN encoder is applied to obtain node representations. After that, GraphMAE re-masks these representations, applies a GNN decoder to reconstruct the features, and uses a scaled cosine error loss.

## 3 EVALUATING GRAPH SELF-SUPERVISED LEARNING METHODS

We begin with a thorough evaluation of representative graph self-supervised learning methods in a unified setup. In this evaluation, we pay special attention to making the results reliable by considering diverse datasets and carefully tuning both supervised baselines and the considered graph SSL methods.

**Supervised baselines** As supervised baselines, we select three classic message-passing neural networks that are frequently used in graph ML studies — GCN (Kipf & Welling, 2016b), Graph-SAGE (Hamilton et al., 2017), and Graph Transformer[2] (GT) (Shi et al., 2020).

As shown by Luo et al. (2024), classic GNNs are often undertuned, and simple architectural enhancements together with hyperparameter search can greatly affect their performance. Thus, we augment all the considered models with residual connections (He et al., 2016), layer normalization (Ba et al., 2016), and dropout (Hinton et al., 2012). We also use a two-layer MLP instead of just a linear layer after each neighborhood aggregation layer. Additionally, we separate ego- and neighbor-embeddings in the neighborhood aggregation layers, similar to the way this is done in GraphSAGE (Hamilton et al., 2017). This separation was shown to be highly efficient for heterophilous datasets (Zhu et al., 2020a; Platonov et al., 2023). Finally, for datasets with tabular features, we transform numerical features to the standard normal distribution via quantile transformation and use `Periodic-Linear-ReLU` (PLR) numerical feature embeddings (Gorishniy et al., 2022) that are known to significantly improve the performance in the presence of tabular feature.

**Self-supervised methods** Following our discussion in Section 2, we select the following representative graph SSL methods for evaluation: GRACE (Zhu et al., 2020b) and BGRL (Thakoor et al., 2022) that are two contrastive methods, along with MaskGAE (Li et al., 2023) and GraphMAE (Hou et al., 2022) that represent generative approaches.

For the evaluation of SSL models on downstream tasks, we employ two strategies. The first one is linear probing, the strategy commonly used in the literature (Zhu et al., 2020b; Thakoor et al., 2022; Hou et al., 2022; Li et al., 2023), where the pre-trained GNN model is frozen, and a linear classifier or regressor is trained on top of the representations of the last layer of the pre-trained model. The second strategy is full finetune, where a linear head replaces the last layer of the pre-trained GNN model, and both the linear head and the GNN are jointly trained on the downstream task.

---

[2]This version of GT uses only local attention to the node's neighbors.

Table 1: The key statistics of the considered graph datasets. In feature type column, 'tabular' denotes heterogeneous numerical and categorical features, 'BoW' denotes bag-of-words representations.

| name | # nodes | # edges | # features | mean degree | # classes | homophily | feature type |
|---|---|---|---|---|---|---|---|
| cora | 2,708 | 10,556 | 1,433 | 3.9 | 7 | yes | BoW |
| citeseer | 3,327 | 9,104 | 3,703 | 2.74 | 6 | yes | BoW |
| amazon-photo | 7,650 | 238,162 | 745 | 31.13 | 8 | yes | BoW |
| amazon-computers | 13,752 | 491,722 | 767 | 35.76 | 10 | yes | BoW |
| pubmed | 19,717 | 88,648 | 500 | 4.5 | 3 | yes | embedding |
| lastfm-asia | 7,624 | 55,612 | 128 | 7.29 | 18 | yes | embedding |
| facebook | 22,470 | 341,646 | 128 | 15.2 | 4 | yes | embedding |
| amazon-ratings | 24,492 | 186,100 | 300 | 7.6 | 5 | no | embedding |
| tolokers-tab | 11,758 | 1,038,000 | 9 | 88.28 | 2 | no | tabular |
| questions-tab | 48,921 | 307,080 | 40 | 6.28 | 2 | no | tabular |

To provide a fair comparison, we re-implement all the considered methods in a unified codebase, ensuring that they have exactly the same evaluation protocol, including dataset splits, implementation details of GNN architectures and linear probing, etc. For SSL methods, we employ the same GNN models as for the supervised baselines.

**Hyperparameter optimization** We optimize the hyperparameters of the considered baselines with Optuna (Akiba et al., 2019). Specifically, we run 100 trials of TPESampler with 10 initial trials and optimize the dropout rate, the number of layers, the dimension of hidden layers, and the learning rate. When using PLR, we also optimize the number of frequencies, the frequency scale, and the embedding dimension. Further details, including hyperparameter search spaces, are described in Appendix A.

We exploit the same hyperparameter optimization for the GNN backbones used by SSL methods as for their standard supervised counterparts. However, various SSL methods have their own hyperparameters that affect the difficulty and effectiveness of the associated pretraining procedure, so we additionally include these hyperparameters in the optimization (see details in Appendix A).

As hyperparameter optimization introduces some additional noise to evaluation, we re-optimize hyperparameters for baselines 10 times and report mean and standard deviation across Optuna restarts. This is a rather expensive procedure, so for SSL methods, we optimize hyperparameters only once. While this can lead to noisy results on some datasets, the overall results across all datasets are still reliable due to the comparison with well-tuned baselines combined with a large number of datasets in our evaluation.

**Datasets** We select the following representative graph datasets: cora, citeseer, and pubmed are three popular citation networks (Giles et al., 1998; McCallum et al., 2000; Sen et al., 2008; Namata et al., 2012; Yang et al., 2016); lastfm and facebook are social networks (Rozemberczki & Sarkar, 2020; Rozemberczki et al., 2021); amazon-photo and amazon-computers are popular co-purchasing networks (Shchur et al., 2018); amazon-ratings is a heterophilous graph dataset from Platonov et al. (2023); tolokers-tab and questions-tab are two datasets from the recently proposed TabGraphs[3] benchmark (Bazhenov et al., 2025) that provides diverse prediction tasks and includes graphs with tabular node features.

The dataset statistics are shown in Table 1. Importantly, our datasets cover both homophilous and heterophilous settings. In homophilous graphs, edges tend to connect similar nodes, i.e., those with the same class label. The opposite of homophily is heterophily, and there is a growing discussion on the necessity of developing specialized models tailored specifically for heterophilous settings (Zhu et al., 2020a; Ma et al., 2022; Luan et al., 2022; Platonov et al., 2023). Among our datasets, amazon-ratings, tolokers-tab and questions-tab are heterophilous, and the remaining ones are homophilous. Our evaluation includes datasets with tabular (heterogeneous) node features typical for practical applications. These datasets were recently proposed by Bazhenov et al. (2025) to draw the attention of the graph machine learning community to more realistic settings.

---

[3]Concurrently with this submission, TabGraphs has been superseded by GraphLand, and we are going to use GraphLand in future revisions. However, in this submission we have used the datasets from TabGraphs.

Table 2: Comparing SSL methods on downstream problems. We use linear probing (LP) or full finetuning (FullFT) adaptation strategies. We report average precision for binary classification and accuracy for multiclass classification. In the last column, we report the average rank (AR) for each method over all datasets but within a specific model (GCN, GraphSAGE, or GT), the smaller the better. For each column and model, we highlight first, second and third best results with a color.

| | Method | cora | citeseer | pubmed | lastfm-as. | facebook | photo | computers | tolok.-tab | quest.-tab | ratings | AR |
|---|---|---|---|---|---|---|---|---|---|---|---|---|
| | | | | | | Results for GCN | | | | | | |
| | GCN | 80.39 ± 0.56 | 71.57 ± 0.57 | 86.47 ± 0.27 | 81.48 ± 0.74 | 92.69 ± 0.14 | 94.10 ± 0.14 | 89.44 ± 0.16 | 53.91 ± 1.96 | 82.80 ± 0.84 | 41.15 ± 0.43 | 7.50 |
| LP | MaskGAE | 50.21 ± 22.55 | 67.72 ± 1.01 | 86.22 ± 0.20 | 85.92 ± 0.22 | 91.09 ± 0.32 | 93.78 ± 0.32 | 89.74 ± 0.21 | 58.94 ± 1.29 | 79.12 ± 3.13 | 41.72 ± 0.51 | 8.20 |
| | GraphMAE | 78.58 ± 0.84 | 70.01 ± 0.58 | 85.32 ± 0.18 | 77.85 ± 0.42 | 86.90 ± 0.37 | 92.88 ± 0.35 | 89.86 ± 0.21 | 56.26 ± 0.82 | 74.21 ± 1.17 | 40.84 ± 0.34 | 9.70 |
| | BGRL | 78.93 ± 1.21 | 69.37 ± 0.48 | 87.71 ± 0.18 | 80.35 ± 0.56 | 90.63 ± 0.25 | 93.90 ± 0.16 | 90.57 ± 0.23 | 60.55 ± 0.56 | 83.41 ± 0.59 | 38.74 ± 0.27 | 7.30 |
| | GRACE | 80.63 ± 1.36 | 71.27 ± 0.58 | 88.12 ± 0.17 | 82.65 ± 0.29 | 91.65 ± 0.15 | 94.41 ± 0.20 | 91.19 ± 0.15 | 55.31 ± 1.27 | 78.83 ± 0.83 | 40.01 ± 0.38 | 6.40 |
| | GrASP | 83.64 ± 0.78 | 71.37 ± 0.36 | 87.68 ± 0.19 | 86.03 ± 0.36 | 92.88 ± 0.17 | 94.70 ± 0.18 | 91.76 ± 0.19 | 60.85 ± 1.14 | 87.66 ± 0.39 | 41.56 ± 1.69 | 2.60 |
| FullFT | MaskGAE | 82.56 ± 0.97 | 70.71 ± 0.68 | 87.49 ± 0.14 | 86.49 ± 0.18 | 92.84 ± 0.19 | 94.55 ± 0.24 | 90.42 ± 0.36 | 55.43 ± 1.27 | 83.20 ± 0.54 | 42.31 ± 0.49 | 4.80 |
| | GraphMAE | 81.04 ± 0.87 | 70.45 ± 0.40 | 87.05 ± 0.17 | 82.94 ± 0.44 | 93.23 ± 0.16 | 94.64 ± 0.21 | 90.56 ± 0.30 | 54.51 ± 1.15 | 81.18 ± 1.21 | 42.65 ± 0.57 | 6.00 |
| | BGRL | 80.32 ± 0.72 | 71.34 ± 0.77 | 87.37 ± 0.23 | 81.99 ± 0.42 | 93.53 ± 0.26 | 94.32 ± 0.20 | 91.17 ± 0.25 | 60.02 ± 0.37 | 83.12 ± 0.79 | 40.70 ± 0.32 | 5.60 |
| | GRACE | 83.51 ± 0.43 | 70.68 ± 1.14 | 88.00 ± 0.16 | 85.77 ± 0.37 | 93.63 ± 0.10 | 94.40 ± 0.19 | 90.98 ± 0.15 | 55.46 ± 2.17 | 81.79 ± 0.64 | 41.42 ± 0.48 | 4.90 |
| | GrASP | 84.23 ± 0.37 | 71.50 ± 0.39 | 88.09 ± 0.22 | 86.02 ± 0.35 | 93.36 ± 0.13 | 94.67 ± 0.27 | 91.46 ± 0.29 | 56.74 ± 1.73 | 81.72 ± 1.67 | 41.90 ± 0.77 | 3.00 |
| | | | | | | Results for GraphSAGE | | | | | | |
| | GraphSAGE | 80.91 ± 0.84 | 70.80 ± 0.76 | 86.02 ± 0.17 | 83.12 ± 0.61 | 93.05 ± 0.20 | 94.26 ± 0.06 | 89.42 ± 0.28 | 56.01 ± 1.07 | 82.03 ± 1.39 | 41.36 ± 0.42 | 8.20 |
| LP | MaskGAE | 78.15 ± 0.77 | 71.37 ± 0.51 | 85.59 ± 0.27 | 86.07 ± 0.36 | 91.92 ± 0.20 | 93.42 ± 0.27 | 89.63 ± 0.24 | 56.16 ± 1.05 | 66.06 ± 32.29 | 40.83 ± 0.39 | 8.70 |
| | GraphMAE | 78.55 ± 1.16 | 69.39 ± 0.83 | 85.35 ± 0.40 | 81.78 ± 0.50 | 88.91 ± 0.21 | 93.81 ± 0.30 | 90.71 ± 0.39 | 56.52 ± 0.95 | 80.17 ± 0.81 | 41.38 ± 0.31 | 8.85 |
| | BGRL | 78.24 ± 1.03 | 67.32 ± 1.68 | 87.17 ± 0.16 | 84.31 ± 0.20 | 91.78 ± 0.16 | 93.26 ± 0.22 | 90.23 ± 0.27 | 60.09 ± 0.44 | 84.61 ± 1.86 | 39.94 ± 0.50 | 7.75 |
| | GRACE | 79.92 ± 0.89 | 70.23 ± 0.73 | 87.79 ± 0.15 | 84.97 ± 0.26 | 93.07 ± 0.09 | 94.51 ± 0.22 | 91.29 ± 0.21 | 58.67 ± 1.35 | 82.96 ± 1.20 | 40.72 ± 0.26 | 5.95 |
| | GrASP | 82.56 ± 0.60 | 72.49 ± 0.71 | 87.14 ± 0.25 | 86.67 ± 0.21 | 93.31 ± 0.18 | 94.72 ± 0.10 | 92.00 ± 0.28 | 60.52 ± 0.36 | 87.29 ± 0.24 | 41.78 ± 0.25 | 2.40 |
| FullFT | MaskGAE | 82.12 ± 0.84 | 69.06 ± 1.02 | 87.15 ± 0.19 | 86.70 ± 0.45 | 93.33 ± 0.18 | 94.58 ± 0.27 | 90.47 ± 0.27 | 58.97 ± 0.96 | 79.97 ± 0.79 | 42.88 ± 0.30 | 5.00 |
| | GraphMAE | 81.72 ± 0.62 | 70.86 ± 0.78 | 87.09 ± 0.21 | 83.58 ± 0.47 | 93.15 ± 0.15 | 94.72 ± 0.21 | 91.29 ± 0.27 | 56.64 ± 1.20 | 79.84 ± 1.60 | 42.36 ± 0.32 | 6.00 |
| | BGRL | 81.11 ± 0.63 | 71.48 ± 0.61 | 87.32 ± 0.12 | 84.04 ± 0.42 | 93.31 ± 0.15 | 94.67 ± 0.23 | 90.71 ± 0.22 | 60.01 ± 0.97 | 84.68 ± 0.71 | 39.94 ± 0.40 | 5.05 |
| | GRACE | 82.15 ± 0.66 | 71.91 ± 0.64 | 87.91 ± 0.21 | 86.36 ± 0.26 | 93.73 ± 0.21 | 94.50 ± 0.20 | 91.64 ± 0.09 | 59.53 ± 0.75 | 83.12 ± 0.65 | 39.15 ± 1.00 | 3.80 |
| | GrASP | 82.00 ± 0.53 | 71.64 ± 0.70 | 88.20 ± 0.25 | 85.96 ± 2.15 | 93.61 ± 0.23 | 94.24 ± 0.16 | 91.50 ± 0.15 | 57.88 ± 1.75 | 80.72 ± 0.87 | 42.01 ± 0.62 | 4.30 |
| | | | | | | Results for GT | | | | | | |
| | GT | 80.99 ± 0.60 | 70.16 ± 0.57 | 85.92 ± 0.22 | 82.83 ± 0.57 | 92.93 ± 0.18 | 94.00 ± 0.23 | 88.80 ± 0.34 | 55.44 ± 3.38 | 81.15 ± 1.69 | 40.89 ± 0.45 | 8.40 |
| LP | MaskGAE | 78.42 ± 1.13 | 70.08 ± 0.98 | 86.78 ± 0.22 | 85.55 ± 0.15 | 91.40 ± 0.21 | 93.35 ± 0.16 | 88.45 ± 0.66 | 58.59 ± 0.79 | 76.01 ± 16.68 | 40.80 ± 0.42 | 8.85 |
| | GraphMAE | 80.48 ± 0.72 | 69.06 ± 0.54 | 85.15 ± 0.29 | 80.87 ± 0.39 | 89.67 ± 0.39 | 93.31 ± 0.29 | 90.34 ± 0.21 | 49.29 ± 1.67 | 80.97 ± 1.23 | 39.96 ± 0.52 | 10.20 |
| | BGRL | 79.46 ± 1.36 | 69.52 ± 1.24 | 86.78 ± 0.20 | 84.36 ± 0.26 | 93.22 ± 0.31 | 94.34 ± 0.14 | 90.52 ± 0.89 | 59.62 ± 0.89 | 84.87 ± 0.60 | 40.20 ± 0.32 | 6.55 |
| | GRACE | 82.25 ± 0.63 | 71.05 ± 0.68 | 87.93 ± 0.23 | 84.08 ± 0.29 | 93.69 ± 0.10 | 94.44 ± 0.19 | 91.29 ± 0.22 | 60.44 ± 1.46 | 84.60 ± 0.53 | 40.49 ± 0.23 | 4.30 |
| | GrASP | 84.06 ± 0.18 | 72.43 ± 0.37 | 87.68 ± 0.45 | 86.67 ± 0.29 | 93.42 ± 0.09 | 95.20 ± 0.22 | 91.68 ± 0.42 | 59.69 ± 0.62 | 86.63 ± 0.32 | 41.70 ± 0.38 | 2.30 |
| FullFT | MaskGAE | 80.60 ± 1.16 | 68.80 ± 0.58 | 87.40 ± 0.30 | 86.07 ± 0.29 | 93.02 ± 0.24 | 94.20 ± 0.27 | 90.70 ± 0.27 | 60.20 ± 2.59 | 83.94 ± 0.62 | 42.94 ± 0.36 | 5.60 |
| | GraphMAE | 81.01 ± 0.99 | 71.15 ± 0.55 | 86.99 ± 0.13 | 84.23 ± 0.55 | 93.74 ± 0.16 | 94.32 ± 0.22 | 90.58 ± 0.09 | 53.64 ± 4.48 | 81.47 ± 1.79 | 42.58 ± 0.31 | 5.90 |
| | BGRL | 80.72 ± 0.78 | 71.84 ± 0.61 | 87.52 ± 0.12 | 85.78 ± 0.29 | 94.02 ± 0.18 | 94.35 ± 0.25 | 89.90 ± 0.32 | 52.16 ± 1.54 | 84.57 ± 0.63 | 40.98 ± 0.49 | 5.20 |
| | GRACE | 81.35 ± 1.16 | 72.19 ± 0.48 | 87.46 ± 0.17 | 86.44 ± 0.42 | 94.35 ± 0.18 | 94.81 ± 0.26 | 90.49 ± 0.33 | 56.23 ± 2.90 | 82.41 ± 1.91 | 40.93 ± 0.41 | 4.30 |
| | GrASP | 83.08 ± 0.91 | 71.83 ± 0.56 | 88.07 ± 0.15 | 86.76 ± 0.38 | 93.91 ± 0.10 | 94.06 ± 0.19 | 90.65 ± 0.28 | 43.01 ± 9.12 | 83.66 ± 0.51 | 42.48 ± 0.37 | 4.40 |

For the datasets with tabular node features, we use the official RL split with 10%/10%/80% proportions for train/validation/test, and for the remaining datasets, we use random stratified splits with the same proportions, following a split ratio often used in graph SSL literature (Zhu et al., 2020b; Thakoor et al., 2022). The labels of the training set are used to train supervised baselines, finetune SSL-pretrained models, or perform linear probing on them. The validation labels are used for early stopping and hyperparameter optimization. We report the performance metrics computed on the test part. Note that we consider a transductive setup, where all nodes in the dataset are known in advance and can be used as model inputs. In particular, during the SSL pretraining stage, we can exploit all the graph nodes as we do not use their labels, relying instead only on the node features and graph structure.

We measure average precision for binary classification and accuracy for multiclass classification. For each dataset, we use one data split that is shared across all methods and all runs. For each method, when optimal hyperparameters are obtained, we re-evaluate it 10 times with different random seeds affecting model initialization and randomness during training (like dropout or masks) and report the corresponding mean and standard deviation of the test metric.[4]

**Results**  The results of the evaluation are presented in Table 2.[5] In addition to the results on each dataset, we also report the average rank (AR) across all datasets but within a specific GNN backbone architecture (GCN, GraphSAGE, or GT). From these results, we can make the following observations.

> **Observation 1.** Contrary to prior literature, GraphMAE and MaskGAE with linear probing on average do not outperform a well-tuned supervised baseline.

Indeed, in the original GraphMAE and MaskGAE papers (Hou et al., 2022; Li et al., 2023) and subsequent studies (Tan et al., 2023; Liang et al., 2025), GraphMAE and MaskGAE were shown to

---

[4]For supervised baselines, mean and standard deviations are reported for different Optuna seeds.

[5]Extremely high values of std are due to divergence of certain experiments. For fair comparison, we do not employ any manual restarts of those experiments.

outperform supervised baselines in most cases. On the contrary, our results show that, with linear probing, these methods are often close to or even underperform the supervised baseline. In particular, GraphMAE and MaskGAE with linear probing have worse AR values compared to the corresponding baseline in Table 2. We argue that this inconsistency with prior works is due to insufficient tuning of supervised baselines in the literature.

> **Observation 2.** GRACE and BGRL with linear probing mostly perform better than supervised baselines, MaskGAE, and GraphMAE.

While it is expected that GRACE and BGRL outperform the supervised baseline, it can be considered surprising that these contrastive learning methods perform better than generative approaches Graph-MAE and MaskGAE. This is notable because generative methods have been shown to achieve better results in other domains like NLP and CV (Devlin et al., 2019; He et al., 2022).

> **Observation 3.** Full finetune can further boost the performance of SSL-pretrained models. In particular, it allows all methods to outperform the supervised baseline on average.

Despite this observation not being surprising, we want to highlight that full finetune is often an easy way to further boost the performance of SSL-pretrained models. While it comes with increased computational cost, we find it negligible, as current graph models are relatively small and fast to train.

Overall, we find it surprising that, under linear probing, GraphMAE and MaskGAE perform worse than both the supervised baseline and contrastive methods. The fact that they are able to surpass the level of supervised baseline, when evaluated with full finetuning, suggests that MaskGAE and GraphMAE still capture some useful information that is not captured by the baseline. However, we hypothesize that this information is not enough. Specifically, one would expect that both graph structure and node features are beneficial for a downstream task. Yet, the MaskGAE objective focuses solely on graph structure, and the GraphMAE objective includes only feature reconstruction. We suppose that focusing on a single source of information does not allow MaskGAE and GraphMAE to capture all the information needed for the downstream task. To verify this hypothesis, in the next section, we conduct an analysis of what information is captured by different models.

## 4 ANALYSIS

In this section, we conduct experiments on synthetic and real data to deepen our understanding of the considered SSL methods, and in particular investigate why some of the methods perform noticeably worse than others (see Section 3). As mentioned above, we hypothesize that poor performance of MaskGAE and GraphMAE can be caused by the fact that these methods capture only a part of the useful signal during the pretraining phase. To support this intuition, we analyze what information about the graph structure and node features can be reconstructed from the learned embeddings of the last layer of a model.

**Probing**   We employ MLP-probing to quantify the quality of reconstruction. Specifically, we get the embeddings from the last layer of a frozen model that was either pretrained on the SSL objective or trained from scratch in a supervised manner. We use these embeddings to train an MLP to predict certain characteristics of a graph.

**Characteristics**   We are interested in whether the embeddings capture information about graph structure and node features. To characterize graph structure, we generate two sets of positive and negative samples. Positive samples are given by node pairs that are connected by an edge in the graph, while negative samples are given by node pairs that are not connected by an edge. An MLP is then trained to classify if the considered sample is negative or positive. For node features, we directly train an MLP to predict them.

Further details on the probing procedure can be found in Appendix B.

**Synthetic datasets**   To create a graph dataset, we adopt the graph structure from some of the real datasets described in Section 3. This is done to obtain realistic and diverse graph structures instead

Table 3: Aggregated results of the reconstruction analysis: the number of graphs (out of 10) such that the model pretrained with the considered SSL method reconstructs the corresponding characteristic better than the supervised baseline. For each graph, we re-run experiment 30 times with different seeds that affect the model, the features and the target. `StrucAug` stands for purely structure-based augmentations, and `FeatAug` refers to augmentations that only affect node features.

| Characteristic | MaskGAE | GraphMAE | BGRL `FeatAug` | BGRL `StrucAug` | GRACE `FeatAug` | GRACE `StrucAug` |
|---|---|---|---|---|---|---|
| Structure | **10/10** | 5/10 | **10/10** | 2/10 | **10/10** | 3/10 |
| Features | 5/10 | **9/10** | 0/10 | **10/10** | 0/10 | **10/10** |

of relying on some random graph generator. To create node features, we first generate a sample from the 16-dimensional standard normal distribution in each node independently of other nodes. We then pass them through a random MLP with the same output dimension to create non-trivial correlations between features. Finally, we introduce correlation between features of neighboring nodes by defining $X_{\text{final}} = \frac{1}{\sqrt{2}} \left( X + D^{-1/2} A X \right)$, where $A$ is the adjacency matrix, $D$ is the degree matrix, and $X$ is the matrix of node features.

The task being solved is node regression, and the target for each node is constructed as follows:

$$\text{target} = \text{target}_{\text{feat}} + \text{target}_{\text{struc}} + \text{target}_{\text{NFA}} + \varepsilon.$$

Here, the $\text{target}_{\text{feat}}$ component depends solely on the node features and is obtained by applying a randomly initialized MLP to these features. We also normalize $\text{target}_{\text{feat}}$ by subtracting its mean and dividing by its standard deviation in order to ensure that the magnitude of different components is the same. The $\text{target}_{\text{struc}}$ component depends only on the graph structure. Namely, inspired by Kanatsoulis et al. (2025), we fix a random GNN with input and output dimensions equal to one, apply it to random inputs multiple times, and take the mean of the outputs as a target. To make the prediction task more suitable for GNNs, we also add the $\text{target}_{\text{NFA}}$ term, which depends on the features of the node neighbors. Specifically, in each node for each feature we compute its mean over the neighboring nodes and apply an MLP to these mean values, similar to the way it is done for the two previous components. We call this term $\text{target}_{\text{NFA}}$ since it is inspired by the neighborhood feature aggregation (NFA) technique from Bazhenov et al. (2025). Finally, $\varepsilon$ represents the noise: it is an i.i.d. sample from the standard normal distribution.

**Models** For this experiment, we selected GT as the GNN backbone for both supervised baseline and SSL methods. Unlike the experiments in Section 3, we did not optimize hyperparameters but instead fixed them to some reasonable values. Importantly, the model and learning hyperparameters were shared between the supervised baseline and SSL methods, ensuring a fair comparison. We used the same SSL methods as in Section 3 but made an important modification to the contrastive methods. Recall that in their default implementation, these methods utilize both feature and structure augmentations. We found that different strength of augmentations can significantly affect the final results. Therefore, we decomposed their augmentations into purely structural and purely feature-based versions. The resulting methods are identified by the suffixes `StructAug` and `FeatAug`, respectively.

**Results** The aggregated results of our reconstruction analysis on synthetic data are shown in Table 3; the full table with individual datasets can be found in Appendix B. From these results, we can make the following observations.

> **Observation 4.** MaskGAE and GraphMAE primarily capture information about a single component, either graph structure or node features.

This observation is rather expected, as the GraphMAE objective is feature reconstruction, and the MaskGAE objective is to reconstruct masked edges and predict node degrees. Thus, these methods mostly capture the information about the characteristic they learn to reconstruct.

Table 4: Downstream metrics of GRACE with different augmentation types on real-world datasets. In this comparison, the GNN backbone is GT for both supervised baseline and SSL method.

| Method | cora | citeseer | pubmed | lastfm-as. | facebook | photo | computers | tolok.-tab | quest.-tab | ratings | AR |
|---|---|---|---|---|---|---|---|---|---|---|---|
| GT | $80.99 \pm 0.60$ | $70.16 \pm 0.57$ | $85.92 \pm 0.22$ | $82.83 \pm 0.57$ | $92.93 \pm 0.18$ | $94.00 \pm 0.23$ | $88.80 \pm 0.34$ | $55.44 \pm 3.38$ | $81.15 \pm 1.69$ | $40.89 \pm 0.45$ | **2.20** |
| GRACE | $82.25 \pm 0.63$ | $71.05 \pm 0.68$ | $87.93 \pm 0.23$ | $84.08 \pm 0.29$ | $93.69 \pm 0.10$ | $94.44 \pm 0.19$ | $91.29 \pm 0.22$ | $60.44 \pm 1.46$ | $84.60 \pm 0.53$ | $40.49 \pm 0.23$ | **1.10** |
| GRACE StrucAug | $75.90 \pm 2.07$ | $70.00 \pm 0.81$ | $84.35 \pm 0.26$ | $81.51 \pm 0.47$ | $87.12 \pm 0.48$ | $94.06 \pm 0.33$ | $89.84 \pm 0.30$ | $58.54 \pm 1.60$ | $81.06 \pm 0.49$ | $38.61 \pm 0.55$ | **2.80** |
| GRACE FeatAug | $79.76 \pm 1.23$ | $69.52 \pm 0.65$ | $83.92 \pm 0.11$ | $80.16 \pm 0.49$ | $87.00 \pm 0.40$ | $92.64 \pm 0.20$ | $84.55 \pm 0.45$ | $53.82 \pm 1.92$ | $74.28 \pm 4.37$ | $36.55 \pm 0.59$ | **3.90** |

Table 5: Ablation results for modifications introduced in GrASP compared to GrASP$_{\text{naive}}$. StrucMod and FeatMod stand for modifications to the structure-based and feature-based components, respectively. GrASP is GrASP$_{\text{naive}}$ with both modifications applied simultaneously.

| Method | cora | citeseer | pubmed | lastfm-as. | facebook | photo | computers | tolok.-tab | quest.-tab | ratings | AR |
|---|---|---|---|---|---|---|---|---|---|---|---|
| GT | $80.99 \pm 0.60$ | $70.16 \pm 0.57$ | $85.92 \pm 0.22$ | $82.83 \pm 0.57$ | $92.93 \pm 0.18$ | $94.00 \pm 0.23$ | $88.80 \pm 0.34$ | $55.44 \pm 3.38$ | $81.15 \pm 1.69$ | $40.89 \pm 0.45$ | **4.10** |
| GrASP$_{\text{naive}}$ | $79.63 \pm 0.76$ | $70.33 \pm 0.50$ | $86.61 \pm 0.22$ | $86.47 \pm 0.27$ | $92.17 \pm 0.10$ | $93.91 \pm 0.23$ | $90.35 \pm 0.16$ | $59.47 \pm 0.92$ | $73.52 \pm 3.20$ | $41.23 \pm 0.29$ | **3.70** |
| GrASP$_{\text{naive}}$ + StrucMod | $80.48 \pm 1.05$ | $71.03 \pm 0.50$ | $85.67 \pm 0.20$ | $86.41 \pm 0.21$ | $91.97 \pm 0.21$ | $94.14 \pm 0.21$ | $90.69 \pm 0.25$ | $60.92 \pm 0.37$ | $58.50 \pm 22.12$ | $41.16 \pm 0.46$ | **3.50** |
| GrASP$_{\text{naive}}$ + FeatMod | $84.47 \pm 0.33$ | $73.09 \pm 0.47$ | $88.18 \pm 0.22$ | $86.34 \pm 0.20$ | $93.04 \pm 0.21$ | $93.78 \pm 0.57$ | $90.42 \pm 0.37$ | $60.94 \pm 0.85$ | $86.80 \pm 0.85$ | $41.60 \pm 0.41$ | **2.10** |
| GrASP | $84.06 \pm 0.18$ | $72.43 \pm 0.37$ | $87.68 \pm 0.45$ | $86.67 \pm 0.29$ | $93.42 \pm 0.09$ | $95.20 \pm 0.22$ | $91.68 \pm 0.42$ | $59.69 \pm 0.62$ | $86.63 \pm 0.32$ | $41.70 \pm 0.30$ | **1.60** |

> **Observation 5.** GRACE and BGRL with structure augmentations capture information about node features, while feature augmentations lead to improved structure reconstruction.

This observation is less obvious, but still rather natural. Indeed, if a certain characteristic is noisy, meaning that it changes across different augmented views, it is natural to expect that the model will use other characteristics to perform contrasting. However, it is important to mention that in our implementation, contrastive methods can have different configurations of augmentation strengths, allowing them to focus on feature augmentations, structure augmentations, or combine them with different proportions. Therefore, we argue that GRACE and BGRL with different hyperparameter configurations can adaptively capture information about graph structure and node features.

The above observations can explain why MaskGAE and GraphMAE have worse performance compared to BGRL and GRACE. To further support this explanation, we conduct additional experiments by considering GRACE with either only structural or only feature augmentations on the real datasets. The results are presented in Table 4, and can be summarized as follows.

> **Observation 6.** Under linear probing, GRACE with either purely structure-based or feature-based augmentations performs worse than both supervised baseline and the implementation with combined augmentations.

## 5 CAPTURING STRUCTURE AND FEATURES SIGNALS

Based on the observations discussed above, we conclude that the performance of different models, especially under linear probing, is closely related to the information they capture, and both graph structure and node features are necessary to achieve good performance on downstream tasks. Therefore, we propose the following recommendation for developing graph SSL methods.

> **Recommendation.** Graph self-supervised learning objectives should be designed to capture both graph structure and node feature signals.

To verify the effectiveness of such an approach, we first consider a method that directly combines structure-based and feature-based generative methods, namely, MaskGAE and GraphMAE. We call this method GrASP$_{\text{naive}}$: Graph Attribute and Structure Prediction, where 'naive' means that we simply employ the sum of MaskGAE and GraphMAE losses with potentially different coefficients, while preserving all their initial design choices. From the results in Table 8, we observe that, unlike MaskGAE and GraphMAE, GrASP$_{\text{naive}}$ is already able to outperform the supervised baseline on average.

In order to simplify GrASP$_{\text{naive}}$ and further improve its performance, we also propose GrASP, which introduces two modifications to GrASP$_{\text{naive}}$:

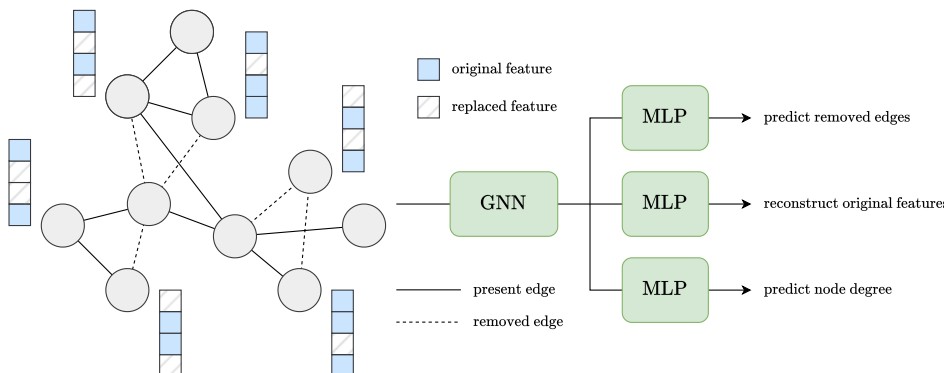

Figure 1: The proposed GrASP method.

- `StrucMod`: For the structure-based component given by MaskGAE, we change Path-wise masking to Edge-wise masking, as Edge-wise masking has fewer hyperparameters and is overall simpler.
- `FeatMod`: For the feature-based component, we propose to use a simpler feature reconstruction objective with a single hyperparameter instead of GraphMAE that has multiple hyperparameters, GNN decoder, re-masking procedure, and employs scaled cosine error. Specifically, our method samples the mask for node features uniformly at random across all distinct features and graph nodes, then replaces the masked entries with other random non-masked values and requires to reconstruct the original values of all features in all nodes using simple MSE loss and MLP decoder.

The proposed framework is illustrated in Figure 1 and operates as follows. First, it randomly masks some edges and node features. Then, both graph structure and node features are jointly processed by a GNN encoder. Finally, representations from the last GNN layer are used in three MLP decoders: the first one reconstructs masked edges, the second one reconstructs the original features, while the third one predicts node degrees. Refer to Appendix C for the discussion of computation complexity. Table 5 presents the ablation results for the modifications described above. The results show that the simplification of the structure-based component does not lead to performance degradation, and sometimes even improves the performance. In turn, the modification in feature-based component not only allows us to simplify the method, but also leads to the improved performance.

Finally, we compare the proposed GrASP method with the other graph SSL methods, with the results presented in Table 2. Despite being surprisingly simple, GrASP outperforms all other considered approaches, and in particular has the best average rank for all the considered GNN backbones.

## 6 CONCLUSION

In this work, we provide a thorough evaluation of representative graph self-supervised learning methods in a unified setup with a focus on node-level tasks. We carefully tune both supervised baselines and graph SSL methods by employing architectural enhancements and a thorough hyperparameter optimization procedure. Our evaluation reveals that, surprisingly, representative generative methods MaskGAE and GraphMAE, on average, perform worse than both the supervised baseline and contrastive approaches. We hypothesize that this is due to the fact that MaskGAE and GraphMAE focus on a single source of information, either graph structure or node features, while contrastive methods BGRL and GRACE are able to capture signals from both sources. We support this hypothesis by conducting analysis on both synthetic and real-world data. Based on our insights, we propose a very simple yet often overlooked recommendation: graph self-supervised learning methods should be designed so that they are able to capture information from both graph structure and node features. We demonstrate the effectiveness of this recommendation by proposing GrASP (Graph Attribute and Structure Prediction), a simple generative method that outperforms all other considered approaches.

**Limitations** First, we intentionally limit the scope of our work to node-level prediction, which is arguably one of the most widespread tasks in graph machine learning literature. This focus allows for a thorough analysis in a unified setup. It would be beneficial to verify the validity of our insights in

other settings, such as link prediction and graph classification. Second, it would also be beneficial to check how our insights transfer to emerging directions in graph self-supervised learning, such as graph foundation models (Zhao et al., 2024a; Xia et al., 2024; Xia & Huang, 2024), in-context learning (Huang et al., 2023) or fully-inductive node classification (Zhao et al., 2025).

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

## A EVALUATION DETAILS

**Supervised baselines**   As mentioned in Section 3, our GNN models are augmented with residual connections, layer normalization, and dropout. They also use two-layer MLP instead of just a linear layer after each neighborhood aggregation layer and separate ego- and neighbor-embeddings during aggregation. Overall, one layer of GCN or GraphSAGE can be described as follows:

$$\text{GCN}: \quad X^{(L+1)} = X^{(L)} + \text{MLP}([\text{LN}(X^{(L)}), D^{-1/2}AD^{-1/2}(\text{LN}(X^{(L)}))]),$$
$$\text{GraphSAGE}: \quad X^{(L+1)} = X^{(L)} + \text{MLP}([\text{LN}(X^{(L)}), D^{-1}A(\text{LN}(X^{(L)}))]),$$

where $X^{(L)}$ corresponds to activations at the $L$'th layer, $A$ is adjacency matrix, $D$ is degree matrix, $[\cdot, \cdot]$ denotes concatenation, and MLP is two-layer MLP with dropout. For GT it is almost the same with the only exception that we employ residual connections both in convolution and in MLP modules:

$$y^{(L)} = x^{(L)} + \text{Linear}([\text{LN}(x^{(L)}), \text{MultiHeadAttn}(\text{LN}(x^{(L)}))]),$$
$$x^{(L+1)} = y^{(L)} + \text{MLP}(\text{LN}(y^{(L)})).$$

We use ReLU activation in all models. In GT, we fix the number of heads to 8. For optimization, we employ Adam (Kingma & Ba, 2014). We use 2500 steps with early stopping on validation metric for learning supervised baselines and finetuning SSL-pretrained models. The number of steps for pretraining is considered as a hyperparameter. When employing finetune of SSL-pretrained models, we separate learning rate for pretrain and finetune stages.

**SSL methods**   We introduce a minor modification to GRACE in order to scale it to larger graphs. Namely, we learn it in a batched manner: at each iteration, we apply the GNN encoder to the whole graph to obtain node representations, but after that, we randomly split all nodes into batches and compute the GRACE loss in each batch independently. In all experiments, we fix the batch size to 2048. For MaskGAE, we employ its MaskGAE$_{\text{path}}$ variant, as it was shown to have better performance. For GraphMAE, we use a single GraphSAGE layer as decoder for all GNN backbones. This GraphSAGE layer employs MLP instead of linear layer, but does not employ dropout, normalization, residual connection and does not separate ego- and neighbor-embeddings.

**Evaluation**   For linear probing, we use the implementation from sklearn (Buitinck et al., 2013). We select regularization parameter from $\{2^{-10}, 2^{-9}, \ldots, 2^{10}\}$ via 5-fold cross-validation on the train part.

**Hyperparameters**   We show the full hyperparameter search spaces in Table D. Optimal hyperparameter configurations for both supervised baselines and considered graph self-supervised learning methods are available in the repository with code.

## B ANALYSIS DETAILS

**Implementation details**   Let us describe the experimental setup that we use in Section 4. We employ Graph Transformer (GT, Shi et al., 2020) architecture for GNN. We use all architectural improvements from Section 3, but do not tune any hyperparameters at all. Importantly, the architectural hyperparameters are shared between the baseline and the SSL-pretrained models, so we get a fair comparison of the reconstruction quality.

For reconstruction, we employ a two-layer MLP with the hidden dimension equal to the doubled dimension of the input. For example, to reconstruct features, we train an MLP with the following (input, hidden, output) layer sizes: (feat_dim, $2 \cdot$ feat_dim, feat_dim). In case of edge prediction, all dimensions except output are the same, while output dimension equals to one. When training MLP to reconstruct characteristics, we employ 3-fold cross-validation. We report ROC-AUC for edge reconstruction and $R^2$ for feature reconstruction.

In order to have more reliable results, we re-evaluate all our models with different random seeds 30 times and report mean and standard deviation over these re-evaluations. Importantly, unlike

Section 3, we generate new split and set of features and targets for each random seed. However, split, features and targets only depend on random seed, but not on considered method, so comparison is fair. Similarly to Section 3, we employ stratified random 10%/10%/80% train/validation/test splits. For the regression tasks on synthetic data, we split the target values into four bins, and use these bins for stratification.

**Additional results**   Table 7 provides full results of the reconstruction analysis that were summarized in Table 3.

## C   COMPUTATIONAL COMPLEXITY OF GRASP

In this section, we elaborate on time and memory complexity of the proposed GrASP method. Let $H$ the hidden dimension, $F$ the number of input features, $E$ the total number of edges, $N$ the number of nodes, and $E_{\text{pos}}$ the number of positive sampled edges. Then, the time complexity for the decoders is $O(NH^2 + NFH + H^2 E_{\text{pos}})$, and the memory requirements are $O(NH + NF + HE_{\text{pos}})$. Overall, GrASP has negligible overhead in terms of time or memory complexity compared to supervised GNN training, as the only extra computations come from the decoders, which are simple MLPs in the case of GrASP (unlike, for example, GraphMAE, which uses GNN as a decoder). The only exception here is the $O(HE_{\text{pos}})$ memory requirement for the decoder, which comes from the fact that we apply the MLP-decoder to all sampled edges. For large dense graphs, this term can lead to an OOM if we sample a fixed ratio of edges. But this can be easily avoided by simply clipping the maximal number of sampled edges, and moreover, this problem is not specific to GrASP and is also relevant to, for example, MaskGAE$_{\text{edge}}$.

## D   LLM USAGE

LLMs have been used for proofreading and minor stylistic editing of the paper; the authors are responsible for all content.

Table 6: Hyperparameter search space. Here, "low" specifies the lower bound, "high" specifies the upper bound, "step" is an optional step size in a grid ("N/A" means that we use the whole interval instead of a discrete grid), "log" indicates whether the space has a logarithmic scale (used, for example, to measure the similarity of different values of the hyperparameter during hyperparameter optimization), and "type" indicates the type of the considered hyperparameter (integer or floating point number).

| | low | high | step | log | type |
|---|---|---|---|---|---|
| General | | | | | |
| dropout | 0.000000 | 0.500000 | 0.100000 | False | float |
| num_layers | 1 | 11 | 1 | False | int |
| hidden_dim | 64 | 1024 | 32 | False | int |
| lr | 0.000100 | 0.020000 | N/A | True | float |
| PLR numerical embeddings | | | | | |
| plr_frequencies_dim | 16 | 96 | 8 | False | int |
| plr_frequencies_scale | 0.010000 | 10.000000 | N/A | True | float |
| plr_embedding_dim | 8 | 32 | 4 | False | int |
| Pretrain general | | | | | |
| pretrain_num_steps | 100 | 10000 | 100 | False | int |
| pretrain_lr | 0.000100 | 0.020000 | N/A | True | float |
| MaskGAE | | | | | |
| coef_degree | 0.000010 | 0.100000 | N/A | True | float |
| anchor_node_prob | 0.200000 | 0.800000 | 0.100000 | False | float |
| num_walks_per_anchor | 1 | 4 | 1 | False | int |
| walk_len | 2 | 8 | 1 | False | int |
| GraphMAE | | | | | |
| masking_rate | 0.500000 | 0.900000 | 0.050000 | False | float |
| replacing_rate | 0.000000 | 1.000000 | 0.050000 | False | float |
| scaling_factor | 1.000000 | 4.000000 | 0.500000 | False | float |
| BGRL | | | | | |
| drop_edge_prob_first | 0.000000 | 0.800000 | 0.100000 | False | float |
| mask_feat_prob_first | 0.000000 | 0.800000 | 0.100000 | False | float |
| drop_edge_prob_second | 0.000000 | 0.800000 | 0.100000 | False | float |
| mask_feat_prob_second | 0.000000 | 0.800000 | 0.100000 | False | float |
| bgrl_ema_decay | 0.980000 | 0.999000 | N/A | True | float |
| GRACE | | | | | |
| drop_edge_prob_first | 0.000000 | 0.800000 | 0.100000 | False | float |
| mask_feat_prob_first | 0.000000 | 0.800000 | 0.100000 | False | float |
| drop_edge_prob_second | 0.000000 | 0.800000 | 0.100000 | False | float |
| mask_feat_prob_second | 0.000000 | 0.800000 | 0.100000 | False | float |
| temperature | 0.010000 | 1.000000 | N/A | True | float |
| GrASP | | | | | |
| coef_degree | 0.000010 | 0.100000 | N/A | True | float |
| edge_mask_prob | 0.200000 | 0.800000 | 0.100000 | False | float |
| feat_corrupt_prob | 0.200000 | 0.800000 | 0.100000 | False | float |
| coef_feat | 0.100000 | 10.000000 | N/A | True | float |

Table 7: Full results of reconstruction quality analysis. For features reconstruction, we report $R^2$, and for edge reconstruction we report ROC-AUC.

| Method | cora | citeseer | pubmed | lastfm-asia | facebook | photo | computers | tolokers-tab | questions-tab | ratings |
|---|---|---|---|---|---|---|---|---|---|---|
| | | | | | features | | | | | |
| Baseline | 92.03 ± 1.50 | 92.52 ± 1.72 | 93.95 ± 0.40 | 93.07 ± 1.33 | 94.01 ± 0.47 | 93.32 ± 1.47 | 94.27 ± 0.44 | 94.32 ± 0.72 | 93.86 ± 0.33 | 94.12 ± 0.40 |
| MaskGAE | 95.31 ± 0.14 | 95.44 ± 0.12 | 95.17 ± 0.11 | 94.49 ± 0.14 | 92.66 ± 0.21 | 89.89 ± 0.26 | 88.84 ± 0.32 | 87.39 ± 0.46 | 95.58 ± 0.15 | 92.57 ± 0.20 |
| GraphMAE | 94.09 ± 0.16 | 94.42 ± 0.17 | 94.37 ± 0.18 | 94.43 ± 0.13 | 94.75 ± 0.15 | 94.39 ± 0.18 | 94.67 ± 0.16 | 93.80 ± 0.26 | 94.31 ± 0.22 | 94.72 ± 0.16 |
| BGRL FeatAug | 80.33 ± 0.84 | 79.48 ± 1.03 | 80.28 ± 1.53 | 80.52 ± 1.14 | 81.58 ± 1.16 | 82.98 ± 1.25 | 83.50 ± 0.85 | 81.75 ± 2.06 | 80.15 ± 1.18 | 81.37 ± 1.46 |
| BGRL StrucAug | 95.66 ± 0.18 | 95.79 ± 0.15 | 96.23 ± 0.25 | 95.86 ± 0.24 | 96.11 ± 0.25 | 95.49 ± 0.16 | 95.59 ± 0.20 | 95.58 ± 0.20 | 96.44 ± 0.18 | 95.98 ± 0.18 |
| GRACE FeatAug | 83.72 ± 0.81 | 84.32 ± 1.12 | 80.88 ± 1.67 | 81.42 ± 1.21 | 81.37 ± 1.27 | 83.98 ± 0.87 | 83.22 ± 0.99 | 81.75 ± 1.47 | 79.10 ± 1.51 | 83.85 ± 1.35 |
| GRACE StrucAug | 95.75 ± 0.39 | 95.81 ± 0.32 | 96.90 ± 0.35 | 96.35 ± 0.38 | 96.80 ± 0.36 | 96.15 ± 0.36 | 96.66 ± 0.35 | 95.95 ± 0.49 | 97.18 ± 0.39 | 96.92 ± 0.33 |
| | | | | | edges | | | | | |
| Baseline | 92.04 ± 0.79 | 95.15 ± 0.59 | 94.05 ± 0.86 | 90.47 ± 1.20 | 90.71 ± 1.10 | 82.84 ± 1.54 | 79.04 ± 1.46 | 81.80 ± 2.26 | 96.37 ± 1.03 | 93.53 ± 0.59 |
| MaskGAE | 99.55 ± 0.08 | 99.58 ± 0.06 | 99.92 ± 0.01 | 99.82 ± 0.03 | 99.93 ± 0.01 | 99.71 ± 0.02 | 99.63 ± 0.02 | 99.02 ± 0.03 | 99.85 ± 0.01 | 99.99 ± 0.00 |
| GraphMAE | 91.36 ± 0.46 | 94.43 ± 0.34 | 92.41 ± 0.57 | 89.05 ± 0.56 | 90.94 ± 0.57 | 86.08 ± 0.95 | 85.55 ± 0.67 | 85.20 ± 0.95 | 97.44 ± 0.19 | 90.72 ± 0.44 |
| BGRL FeatAug | 95.46 ± 0.58 | 97.48 ± 0.33 | 96.70 ± 0.38 | 94.94 ± 0.48 | 94.64 ± 0.46 | 93.09 ± 0.89 | 90.81 ± 1.35 | 92.90 ± 0.78 | 98.09 ± 0.19 | 95.85 ± 0.54 |
| BGRL StrucAug | 89.66 ± 0.52 | 92.61 ± 0.52 | 87.81 ± 3.01 | 85.01 ± 1.79 | 87.04 ± 1.01 | 86.56 ± 0.78 | 82.42 ± 1.53 | 81.70 ± 1.57 | 94.33 ± 1.50 | 91.75 ± 0.37 |
| GRACE FeatAug | 96.71 ± 0.25 | 98.22 ± 0.18 | 98.41 ± 0.10 | 97.15 ± 0.29 | 97.26 ± 0.26 | 97.17 ± 0.33 | 96.41 ± 0.31 | 96.53 ± 0.27 | 99.07 ± 0.04 | 97.32 ± 0.33 |
| GRACE StrucAug | 88.48 ± 0.76 | 91.20 ± 0.71 | 80.89 ± 0.47 | 80.94 ± 0.89 | 78.35 ± 0.78 | 88.01 ± 1.01 | 80.79 ± 5.64 | 85.44 ± 1.63 | 73.44 ± 0.99 | 88.20 ± 0.43 |

Table 8: Comparison of GrASP$_{\text{naive}}$ with other representative graph SSL methods.

| | Method | cora | citeseer | pubmed | lastfm-as. | facebook | photo | computers | tolok.-tab | quest.-tab | ratings | AR |
|---|---|---|---|---|---|---|---|---|---|---|---|---|
| | | | | | | Results for GCN | | | | | | |
| | GCN | 80.39 ± 0.56 | 71.57 ± 0.57 | 86.47 ± 0.27 | 81.48 ± 0.74 | 92.69 ± 0.14 | 94.10 ± 0.14 | 89.44 ± 0.16 | 53.91 ± 1.96 | 82.80 ± 0.84 | 41.15 ± 0.43 | **3.10** |
| LP | MaskGAE | 50.21 ± 22.55 | 67.72 ± 1.01 | 86.22 ± 0.20 | 85.92 ± 0.22 | 91.09 ± 0.32 | 93.78 ± 0.32 | 89.74 ± 0.21 | 58.94 ± 1.29 | 79.12 ± 3.13 | 41.72 ± 0.51 | **4.00** |
| | GraphMAE | 78.58 ± 0.84 | 70.01 ± 0.58 | 85.32 ± 0.18 | 77.85 ± 0.42 | 86.90 ± 0.37 | 92.88 ± 0.35 | 89.86 ± 0.21 | 56.26 ± 0.82 | 74.21 ± 1.17 | 40.84 ± 0.34 | **5.10** |
| | BGRL | 78.93 ± 1.21 | 69.37 ± 0.48 | 87.71 ± 0.18 | 80.35 ± 0.16 | 90.63 ± 0.25 | 93.90 ± 0.16 | 90.57 ± 0.23 | 60.55 ± 0.56 | 83.41 ± 0.59 | 38.74 ± 0.27 | **3.50** |
| | GRACE | 80.63 ± 1.36 | 71.27 ± 0.58 | 88.12 ± 0.17 | 82.65 ± 0.29 | 91.65 ± 0.15 | 94.41 ± 0.20 | 91.19 ± 0.15 | 55.31 ± 1.27 | 78.83 ± 0.83 | 40.01 ± 0.38 | **2.80** |
| | GrASP$_{\text{naive}}$ | 81.39 ± 0.77 | 70.71 ± 0.25 | 85.39 ± 0.27 | 86.22 ± 0.23 | 91.97 ± 0.07 | 94.01 ± 0.33 | 90.40 ± 0.19 | 58.69 ± 0.68 | 79.22 ± 1.44 | 42.09 ± 0.32 | **2.50** |
| | | | | | | Results for GraphSAGE | | | | | | |
| | GraphSAGE | 80.91 ± 0.84 | 70.80 ± 0.76 | 86.02 ± 0.17 | 83.12 ± 0.61 | 93.05 ± 0.20 | 94.26 ± 0.06 | 89.42 ± 0.28 | 56.01 ± 1.07 | 82.03 ± 1.39 | 41.36 ± 0.42 | **3.30** |
| LP | MaskGAE | 78.15 ± 0.77 | 71.37 ± 0.51 | 85.59 ± 0.27 | 86.07 ± 0.36 | 91.92 ± 0.20 | 93.42 ± 0.27 | 89.63 ± 0.24 | 56.16 ± 1.05 | 66.06 ± 32.29 | 40.83 ± 0.39 | **4.00** |
| | GraphMAE | 78.55 ± 1.16 | 69.39 ± 0.83 | 85.35 ± 0.40 | 81.78 ± 0.50 | 88.91 ± 0.21 | 93.81 ± 0.30 | 90.71 ± 0.39 | 56.52 ± 0.95 | 80.17 ± 0.81 | 41.38 ± 0.31 | **4.00** |
| | BGRL | 78.24 ± 1.03 | 67.32 ± 1.68 | 87.17 ± 0.16 | 84.31 ± 0.20 | 91.78 ± 0.16 | 93.26 ± 0.22 | 90.23 ± 0.27 | 60.09 ± 0.44 | 84.61 ± 1.86 | 39.94 ± 0.50 | **3.90** |
| | GRACE | 79.92 ± 0.89 | 70.23 ± 0.73 | 87.79 ± 0.15 | 84.97 ± 0.26 | 93.07 ± 0.09 | 94.51 ± 0.22 | 91.29 ± 0.21 | 58.67 ± 1.35 | 82.96 ± 1.20 | 40.72 ± 0.26 | **2.30** |
| | GrASP$_{\text{naive}}$ | 81.60 ± 0.74 | 70.59 ± 0.51 | 84.71 ± 0.29 | 86.32 ± 0.17 | 92.68 ± 0.13 | 93.00 ± 0.26 | 90.69 ± 0.12 | 55.24 ± 0.93 | 78.52 ± 1.32 | 41.63 ± 0.40 | **3.50** |
| | | | | | | Results for GT | | | | | | |
| | GT | 80.99 ± 0.60 | 70.16 ± 0.57 | 85.92 ± 0.22 | 82.83 ± 0.57 | 92.93 ± 0.18 | 94.00 ± 0.23 | 88.80 ± 0.34 | 55.44 ± 3.38 | 81.15 ± 1.69 | 40.89 ± 0.45 | **3.60** |
| LP | MaskGAE | 78.42 ± 1.13 | 70.08 ± 0.98 | 86.78 ± 0.24 | 85.55 ± 0.15 | 91.40 ± 0.21 | 93.35 ± 0.16 | 88.45 ± 0.66 | 58.59 ± 0.79 | 76.01 ± 16.68 | 40.80 ± 0.42 | **4.25** |
| | GraphMAE | 80.48 ± 0.72 | 69.06 ± 0.54 | 85.15 ± 0.29 | 80.87 ± 0.39 | 89.67 ± 0.39 | 93.31 ± 0.29 | 90.34 ± 0.21 | 49.29 ± 1.67 | 80.97 ± 1.23 | 39.96 ± 0.52 | **5.30** |
| | BGRL | 79.46 ± 1.36 | 69.52 ± 1.24 | 86.78 ± 0.20 | 84.36 ± 0.26 | 93.22 ± 0.31 | 94.34 ± 0.14 | 90.94 ± 0.25 | 59.62 ± 0.89 | 84.87 ± 0.60 | 40.20 ± 0.32 | **2.95** |
| | GRACE | 82.25 ± 0.63 | 71.05 ± 0.68 | 87.93 ± 0.23 | 84.08 ± 0.29 | 93.69 ± 0.10 | 94.44 ± 0.19 | 91.29 ± 0.22 | 60.44 ± 1.46 | 84.60 ± 0.53 | 40.49 ± 0.23 | **1.70** |
| | GrASP$_{\text{naive}}$ | 79.63 ± 0.76 | 70.33 ± 0.50 | 86.61 ± 0.22 | 86.47 ± 0.27 | 92.17 ± 0.10 | 93.91 ± 0.23 | 90.35 ± 0.16 | 59.47 ± 0.92 | 73.52 ± 3.20 | 41.23 ± 0.29 | **3.20** |

