# OpenReview forum: "Capturing Structure and Feature Signals in Graph Self-Supervised Learning"
_ICLR.cc/2026/Conference — Submitted to ICLR 2026_

### Official Review · Reviewer_fb8h · 2025-10-24

**Soundness:** 3
**Presentation:** 3
**Contribution:** 1
**Rating:** 4
**Confidence:** 4

**Summary:**

This paper primarily investigates methods for node-level prediction tasks in Graph SSL. Initially, the paper conducts a thorough evaluation of several representative Graph SSL methods, revealing a surprising finding: the performance of two popular generative methods, MaskGAE and GraphMAE, often fails to surpass that of carefully tuned supervised baselines. Meanwhile, the comparative methods BGRL and GRACE exhibit superior average performance compared to both generative methods and supervised baselines. The authors hypothesize that this is because BGRL and GRACE are capable of simultaneously capturing information about graph structure and node features. Based on these observations, the authors propose a method named grasp, which jointly processes graph structure and node features through a GNN encoder, and then employs three MLP decoders to reconstruct masked edges, reconstruct original features, and predict node degrees, respectively.

**Strengths:**

1. Rigorous benchmarking. The most notable advantage of the paper lies in its comprehensive and rigorous empirical evaluation. The authors conducted experiments on 10 diverse datasets, encompassing homogeneous graphs, heterogeneous graphs, and various types of node features.
2. Fair tuning of the baseline: Unlike previous studies, this paper conducts a thorough hyperparameter search and architectural enhancement for supervised baselines. It is this rigor that reveals that the performance of SSL methods in previous studies may have been overestimated.
3. Clear motivation and concise method: Based on the analysis, the suggestion of "capturing both structural and feature signals" is clear in motivation and instructive.

**Weaknesses:**

1. Limited innovation; this paper is more like an experimental report, which can provide some insight to researchers in graph SSL, but lacks theoretical and methodological innovation. The proposed method, GRASP, merely integrates several existing SSL tasks, which should be common in previous work and represents a relatively trivial innovation in methodology.

2. Scope limited to node-level tasks: The author explicitly states in the limitations section that this study is entirely focused on node-level prediction tasks.
3. Tuning limitations of SSL methods: Although the authors emphasize the importance of tuning, they also acknowledge that due to the high computational cost, the supervised baseline was re-optimized 10 times, while the hyperparameters of the SSL method were only optimized once. This constitutes a potential weakness in the evaluation.

**Questions:**

no

---

> ### Author Response · Authors · 2025-11-24
>
> Thank you for your review and for acknowledging the rigour of our benchmarking and baseline evaluation. We address your concerns below.
>
> > **W1** Limited innovation; this paper is more like an experimental report, which can provide some insight to researchers in graph SSL, but lacks theoretical and methodological innovation. The proposed method, GRASP, merely integrates several existing SSL tasks, which should be common in previous work and represents a relatively trivial innovation in methodology.
>
> First, we fully agree that the proposed GraSP approach is simple and is based on previously known building blocks. However, we consider this to be an important advantage of our work. Indeed, such an algorithm is effective and easy to implement, thus it can be more widespreadly used both by practitioners and as a strong baseline in future academic research. We believe that the graph ML community would benefit from a simple and not over-engineered approach.
>
> Second, our contribution is not just the GrASP method itself, but also the analysis that has led us to proposing it. To the best of our knowledge, the analysis we conducted (Section 4) was not previously presented in the literature on graph self-supervised learning. It gives some new insights into SSL methods, and we believe it may help to design better methods in the future.
>
> > **W2** Scope limited to node-level tasks: The author explicitly states in the limitations section that this study is entirely focused on node-level prediction tasks.
>
> We intentionally focused our study on node-level tasks because link prediction and graph classification present distinct challenges that make it difficult to draw conclusions that apply to all three tasks. Our primary finding is that, for node-level tasks, it is often essential to capture information about both graph structure and node features. However, for link prediction, the graph structure alone can be significantly more important. However, we were able to conduct some experiments on link prediction following the MaskGAE setup. Results are presented in the table below. They show that despite being designed for node-level tasks, GrASP can also be applied to link-level tasks.
>
> | Method | cora | citeseer | pubmed | lastfm-as. | facebook | photo | computers | tolok.-tab | quest.-tab | ratings |
> | ------- | ------------ | ------------ | ------------ | ------------ | ------------ | ------------ | ------------ | ------------ | ------------ | ------------ |
> | Sup GCN | 94.71 ± 0.33 | 91.92 ± 0.72 | 98.73 ± 0.08 | 97.64 ± 0.08 | 99.40 ± 0.01 | 99.36 ± 0.01 | 99.27 ± 0.01 | 98.66 ± 0.20 | 98.61 ± 0.01 | 99.90 ± 0.02 |
> | GrASP | 96.93 ± 0.11 | 96.74 ± 0.20 | 99.11 ± 0.02 | 98.06 ± 0.02 | 99.34 ± 0.02 | 99.42 ± 0.01 | 99.28 ± 0.01 | 98.30 ± 0.16 | 98.51 ± 0.03 | 99.93 ± 0.01 |
>
> Regarding graph classification, the majority of these tasks are found in the molecular, chemical, or biological domains. In such domains, standard techniques like edge masking, which are popular in node-level GSSL methods, may result in graphs that no longer make sense. For this reason, we argue that dedicated, task-specific approaches are required for graph classification, which falls outside the scope of this work.
>
> > **W3** Tuning limitations of SSL methods: Although the authors emphasize the importance of tuning, they also acknowledge that due to the high computational cost, the supervised baseline was re-optimized 10 times, while the hyperparameters of the SSL method were only optimized once. This constitutes a potential weakness in the evaluation.
>
> Let us note that it is a standard practice to tune the parameters of an algorithm once and then report the standard deviation w.r.t. different random seeds that affect model initialization and randomness during training. In addition to that, we conducted a more expensive procedure for the baselines by repeating the parameter optimization to also take into account the randomness of this step. This makes our comparison more reliable compared to the typically used procedure.
>
> Please let us know if you have any additional concerns or questions, we are happy to discuss further.

---

### Official Review · Reviewer_CiDd · 2025-10-28

**Soundness:** 3
**Presentation:** 2
**Contribution:** 2
**Rating:** 4
**Confidence:** 3

**Summary:**

The authors systematically examine the effectiveness of Graph Self-Supervised Learning (GraphSSL) on node-level tasks, with a specific comparison between generative (e.g., MaskGAE, GraphMAE) and contrastive (e.g., GRACE, BGRL) paradigms. They hypothesize that superior performance is closely related to whether a model can simultaneously capture both "structural signals and feature signals." Based on this, the authors further propose a new method, GrASP, and validate its effectiveness on multiple datasets.

**Strengths:**

The paper conducts extensive experiments on both homophilic and heterophilic graph data, covering different types of graph structures and node features, which enhances the generalizability of the conclusions. Furthermore, the proposed GrASP method is conceptually and implementation-wise relatively simple, yet achieves performance improvements across several benchmark tasks, demonstrating certain practical value.

**Weaknesses:**

1. The analysis in the article largely relies on experimental results and lacks theoretical exploration into why simultaneously capturing structure and features is more effective.

2.  Previous research [1] has shown that simple baselines can achieve strong performance with sufficient hyperparameter tuning. How can the authors ensure that the re-run baselines used for comparison with GrASP were indeed sufficiently tuned? According to Table 6 provided in Appendix A, it seems the hyperparameter search space for some models might have missed their optimal settings. For instance, the optimal mask rate for GraphMAE on Cora is reportedly 0.75, but the authors only searched within [0.5, 0.9, 0.05]. Such settings raise concerns about the reliability of the baseline results and the true source of GrASP's improvements.


[1] Classical GNNs are Strong Baselines: Reassessing GNNs for Node Classification. NeurIPS 2024.

**Questions:**

1. For GraphMAE and MaskGAE, their official implementations often use GAT as a strong backbone. Why did the article not include comparisons using a GAT base?

2. In Table 2, GrASP seems to perform notably better on tabular (feature-focused) data. Is there a deeper analysis for this observation?

---

> ### Author Response · Authors · 2025-11-24
>
> Thank you for your detailed review! We address your questions and concerns below.
>
> > **W1** The analysis in the article largely relies on experimental results and lacks theoretical exploration into why simultaneously capturing structure and features is more effective.
>
> Our analysis is indeed empirical. While theoretical analysis seems to be very challenging, we believe that our carefully designed experiments on real and synthetic data give some new insights into SSL methods, and we believe it may help to design better methods in the future.
>
> > **W2** Previous research [1] has shown that simple baselines can achieve strong performance with sufficient hyperparameter tuning. How can the authors ensure that the re-run baselines used for comparison with GrASP were indeed sufficiently tuned? According to Table 6 provided in Appendix A, it seems the hyperparameter search space for some models might have missed their optimal settings. For instance, the optimal mask rate for GraphMAE on Cora is reportedly 0.75, but the authors only searched within [0.5, 0.9, 0.05]. Such settings raise concerns about the reliability of the baseline results and the true source of GrASP's improvements.
>
> Thank you for this comment! We actually find the “Classic GNNs are Strong Baselines” paper to be highly valuable and agree that a proper hyperparameter tuning is an important component of fair comparison. That’s why we employed an extensive hyperparameter search for all considered methods. And we want to clarify that in Table 6 we provide lower and upper bounds of search space, together with an optional step of the grid. So, in the raised example, a mask rate is given by a grid from $0.5$ to $0.9$ with a step $0.05$. That means that the grid is equal to $\\{0.5, 0.55, 0.6, \\ldots, 0.85, 0.9\\}$, and $0.75$ falls perfectly into it. For clarity, we have updated the caption of Table 6 in the new revision.
>
> > **Q1** For GraphMAE and MaskGAE, their official implementations often use GAT as a strong backbone. Why did the article not include comparisons using a GAT base?
>
> Instead of GAT, we use GT, which is a message-passing neural network with the classic Transformer’s multi-head attention. We believe that GT is conceptually similar to GAT, but we can add results with GAT in the camera-ready revision.
>
> > **Q2** In Table 2, GrASP seems to perform notably better on tabular (feature-focused) data. Is there a deeper analysis for this observation?
>
> We have several hypotheses about why this could happen. First, previous approaches may be more suitable for embedding-like features (such as BoW, TF-IDF, Word2Vec, and others) and less suitable for tabular features (for example, consider GraphMAE’s scaled cosine error). Second, existing graph SSL methods may be implicitly overfitted to classic datasets and may perform worse on recently proposed datasets with tabular features.
>
> Please let us know if you have any additional concerns or questions, we are happy to discuss further.

---

### Official Review · Reviewer_4GfW · 2025-10-29

**Soundness:** 3
**Presentation:** 3
**Contribution:** 2
**Rating:** 4
**Confidence:** 3

**Summary:**

This paper investigates graph self-supervised learning (GSSL) and argues that most existing methods capture either structure or feature information, but rarely both, first performing a systematic benchmark of representative GSSL methods  under a unified setup, including GRACE, BGRL, GraphMAE, and MaskGAE.

It shows contrastive methods tend to capture both structure and feature signals, while generative methods often focus on one type only.

Based on these insights, GrASP (Graph Attribute and Structure Prediction) is proposed, a simple generative framework that jointly reconstructs masked edges and node attributes, plus an auxiliary degree-prediction task.
GrASP achieves state-of-the-art performance across ten benchmark datasets while being simpler and more stable than prior generative GSSL approaches.

**Strengths:**

1.	Comprehensive empirical study of major GSSL families (contrastive vs generative) under consistent evaluation.
	2.	Insightful analysis revealing that the type of signal captured (structure vs feature) explains most performance gaps.
	3.	Proposed GrASP framework — a minimal joint prediction objective that unifies structure and feature reconstruction.

**Weaknesses:**

1. Lack of Large-Scale Benchmarks:  All core results are on moderate-scale node-classification datasets (Cora/Citeseer/Pubmed, LastFM, Facebook, Amazon-Photo/Computers, Tolokers, Questions, Ratings). While the set is diverse (homophily/heterophily; tabular vs. homogeneous features), it does not include truly large graphs typical in production or recent GSSL scaling studies. This makes it hard to assess scalability, stability, and efficiency of GrASP and the baselines under realistic constraints (GPU memory pressure, neighbor sampling variance, long training horizons) and to validate the paper’s claims about simplicity vs. performance at scale. The paper itself frames evaluation around node-level tasks with careful tuning, but within a transductive setup and the above dataset suite. Further, I suggest some large-scale dataset like ogbn-products (≈2.4M nodes / 61M edges) as standard transductive node classification and widely used as a “large but manageable” benchmark. Without a large-scale OGB evaluation, external validity remains uncertain.

2. Missing Comparison with Recent Graph SSL Models: The paper benchmarks classic GSSL methods such as GRACE, BGRL, GraphMAE, and MaskGAE, which are indeed canonical baselines. However, the field has recently shifted toward foundation-style graph representation learning, characterized by multi-modal pretraining, large-scale datasets, and Transformer-based architectures. These models, including GraphMVP, GraphMAE-2, GraphGPT and GraphFM / GROOV / UniGraph (2024–2025), aim to unify graph self-supervision under scalable, cross-domain objectives. Without comparison to such models, it remains unclear whether GrASP’s observed simplicity–performance advantage extends beyond the classical GNN-encoder regime. If these recent models cannot be evaluated, please illustrate the reason.

3. The paper’s central claim is that methods that jointly capture structural and feature signals outperform those that focus on a single source—is supported empirically but lacks a formal account of why and when this principle should hold. The current narrative connects performance gaps to what information a method “captures,” based on synthetic and real-data analyses, and then proposes GrASP as a simple joint-reconstruction objective. While convincing in practice, the argument remains predominantly empirical. A theoretical lens would clarify conditions under which joint structure feature pretraining yields provable benefits (e.g., linear probing guarantees, sample complexity improvements), and when it may not.

**Questions:**

See weakness.

---

> ### Author Response · Authors · 2025-11-24
> **Official Comment (Part 1/2)**
>
> Thank you for your review! We address your concerns below.
>
> > **W1**. Lack of Large-Scale Benchmarks: All core results are on moderate-scale node-classification datasets (Cora/Citeseer/Pubmed, LastFM, Facebook, Amazon-Photo/Computers, Tolokers, Questions, Ratings). While the set is diverse (homophily/heterophily; tabular vs. homogeneous features), it does not include truly large graphs typical in production or recent GSSL scaling studies. This makes it hard to assess scalability, stability, and efficiency of GrASP and the baselines under realistic constraints (GPU memory pressure, neighbor sampling variance, long training horizons) and to validate the paper’s claims about simplicity vs. performance at scale. The paper itself frames evaluation around node-level tasks with careful tuning, but within a transductive setup and the above dataset suite. Further, I suggest some large-scale dataset like ogbn-products (≈2.4M nodes / 61M edges) as standard transductive node classification and widely used as a “large but manageable” benchmark. Without a large-scale OGB evaluation, external validity remains uncertain.
>
> In this work, our goal was to rigorously compare well-tuned SSL methods. In particular, we believe a proper hyperparameter tuning to be one of key components of our work. However, straightforwardly applying our evaluation setup to large-scale graphs resulted in prohibitively long experiments. We acknowledge that without such comparison our claims about SOTA performance are less grounded, so we have smoothened them to “GrASP outperforms all considered methods” in the new revision. Please note that our response does not mean that the considered methods cannot be applied to larger graphs, we only claim that it is expensive to conduct a fair comparison without compromising the rigour of our benchmarking, which we believe is crucial for our work and which is also acknowledged by the other reviewers.
>
> Also, we believe that benchmarking GSSL methods on truly large-scale graphs may pose unique challenges. For example, we see a potential pitfall when combining node-wise sampling methods (similar to GraphSAGE sampling) with GSSL methods that use edge masking. Indeed, if the sampled subgraph has a nearly-tree structure, masking edges can lead to significant disconnection of a subgraph, potentially causing degraded performance. Therefore, we believe that an in-depth investigation of graph SSL in large-scale graphs can be a subject of a separate valuable study.
>
> Yet, we believe that our key observation regarding the importance of capturing both graph structure and node features, remains valid and beneficial to future works.
>
> > **W2**. Missing Comparison with Recent Graph SSL Models: The paper benchmarks classic GSSL methods such as GRACE, BGRL, GraphMAE, and MaskGAE, which are indeed canonical baselines. However, the field has recently shifted toward foundation-style graph representation learning, characterized by multi-modal pretraining, large-scale datasets, and Transformer-based architectures. These models, including GraphMVP, GraphMAE-2, GraphGPT and GraphFM / GROOV / UniGraph (2024–2025), aim to unify graph self-supervision under scalable, cross-domain objectives. Without comparison to such models, it remains unclear whether GrASP’s observed simplicity–performance advantage extends beyond the classical GNN-encoder regime. If these recent models cannot be evaluated, please illustrate the reason.
>
> Similar to the previous point, we note that the goal of this work was to rigorously evaluate well-tuned GSSL methods under a unified protocol. It requires us to carefully re-implement all the considered methods, so we retain a relatively small collection of methods. We acknowledge that, without a broader comparison, our claims about SOTA performance are less grounded, so we have smoothened them to “GrASP outperforms all considered methods” in the new revision. However, we also argue that the collection of methods used in our work is representative, and considered methods often appear in other studies and achieve strong performance there [1, 2, 3].
>
> Regarding the mentioned methods, to the best of our knowledge,
> * GraphMVP is only suitable for molecular tasks;
> * GraphMAE-2 does not have better downstream performance on node-level tasks according to [3], so we decided to use the more popular and classic GraphMAE instead;
> * In GraphGPT, SSL method seems to be closely connected with the proposed architecture, and we do not see how to directly apply it to MPNNs. Moreover, it seems like the model performs best on graph- and link-level tasks, while our focus is on node-level tasks;
> * GraphFM does not have open-sourced weights or code;
> * UniGraph is only suitable for text-attributed graphs.
>
> For GROOV, could you please provide a reference since we were not able to find it?

---

> > ### Author Response · Authors · 2025-11-24
> > **Official Comment (Part 2/2)**
> >
> > > **W3**. The paper’s central claim is that methods that jointly capture structural and feature signals outperform those that focus on a single source—is supported empirically but lacks a formal account of why and when this principle should hold. The current narrative connects performance gaps to what information a method “captures,” based on synthetic and real-data analyses, and then proposes GrASP as a simple joint-reconstruction objective. While convincing in practice, the argument remains predominantly empirical. A theoretical lens would clarify conditions under which joint structure feature pretraining yields provable benefits (e.g., linear probing guarantees, sample complexity improvements), and when it may not.
> >
> > While having theoretical exploration can be beneficial, we believe that conducting such analysis under realistic assumptions can be challenging: realistic datasets may have complex graph structures, features of different nature with complex dependences and non-trivial interactions between graph structure and node features. To obtain a tractable setup, one has to oversimplify the problem.
> >
> > We believe that our empirical results are convincing, give some new insights into SSL methods, and may help to design better methods in the future.
> >
> > Please let us know if you have any additional concerns or questions, we are happy to discuss further.
> >
> >
> > ***
> >
> > [1] Tan, Qiaoyu, et al. "S2gae: Self-supervised graph autoencoders are generalizable learners with graph masking." Proceedings of the sixteenth ACM international conference on web search and data mining. 2023.
> >
> > [2] Wang, Yuxiang, et al. "Generative and contrastive paradigms are complementary for graph self-supervised learning." 2024 IEEE 40th International Conference on Data Engineering (ICDE). IEEE, 2024.
> >
> > [3] Zhao, Ziwen, et al. "Masked graph autoencoder with non-discrete bandwidths." Proceedings of the ACM Web Conference. 2024.

---

### Author Response · Authors · 2025-12-03
**Rebuttal summary**

Dear Reviewers and Area Chair,

We would like to thank the reviewers for their valuable feedback and effort in reviewing our submission. Unfortunately, the current rules do not allow the reviewers to continue the discussion. However, we believe to have addressed reviewers' concerns. Below, we summarize the reviews and our responses.

**Strengths:**
- **Rigorous evaluation:**
	- `4GfW`: "Comprehensive empirical study of major GSSL families (contrastive vs generative) under consistent evaluation"
	- `CiDd`: "The paper conducts extensive experiments on both homophilic and heterophilic graph data, covering different types of graph structures and node features, which enhances the generalizability of the conclusions"
	- `fb8h`: "The most notable advantage of the paper lies in its comprehensive and rigorous empirical evaluation", "Unlike previous studies, this paper conducts a thorough hyperparameter search and architectural enhancement for supervised baselines"
- **GrASP simplicity:**
	- `CiDd`: "the proposed GrASP method is conceptually and implementation-wise relatively simple"
	- `fb8h`: "Clear motivation and concise method"
- **Analysis:**
	- `4GfW`: "Insightful analysis revealing that the type of signal captured (structure vs feature) explains most performance gaps"

**Weaknesses:**
- **Limited novelty:** First, we consider the simplicity of GrASP to be its strength, not a weakness. Second, the contribution of our work is not only GrASP itself, but also comprehensive evaluation and analysis.
- **Scope limited to node-level tasks:** We intentionally focused our study on node-level tasks because link prediction and graph classification present distinct challenges that make it difficult to draw conclusions that apply to all three tasks.
- **Lack of large-scale benchmarks and recent baselines:** Following this concern, we adjusted the claims in our paper to clearly reflect the scope of our experiments. We also argue that the collection of methods used in our work is representative, and considered methods often appear in other studies and achieve strong performance there. Finally, we believe that the contribution of our work is not only GrASP itself, but also analysis and simple yet overlooked recommendation for future graph SSL methods.
- **Lack of theoretical exploration:** We acknowledge that our work is fully empirical. However, given ongoing challenges with benchmarking in graph ML [1], we feel that our rigorous evaluation itself offers valuable insights.


Best regards,

Authors

***

[1] Position: Graph Learning Will Lose Relevance Due To Poor Benchmarks, ICML 2025

---

### Meta-Review · Area_Chair_w19E · 2025-12-29

**Summary:**

In this paper, the authors study graph self-supervised learning by providing a hypothesis for the different behaviors of generative methods (such as MaskGAE, GraphMAE) and contrastive methods (such as BGRL and GRACE). Motivated by the observation, they further propose a generative method that reconstructs both graph structures and features.

Though reviewers acknowledge the paper provides meaningful analyses, they raise valid concerns regarding technical novelty (e.g., no theoretical support and the proposed method is too simple) and limited evaluations (e.g., experiments on large-scale benchmarks and more comprehensive baselines). The authors provide a rebuttal, but the reviewers did not (get to) respond. Overall, I think the paper provides an interesting perspective for graph self-supervised learning, but in its current form, the novelty and experimental depth, as all reviewers pointed out, are not sufficient to meet the acceptance bar and thus recommend rejection.

**Reviewer Concerns:**

Reviewer 4GfW:
W1-Lack of Large-Scale Benchmarks: likely not addressed.
W2-Missing Comparison with Recent Graph SSL Model: likely not addressed.
W3-Formal account of why and when this principle should hold: likely not addressed.

Reviewer CiDd:
W1-lacks theoretical exploration: likely not addressed.
W2-hyper-paramter tuning: partially addressed.

Reviewer fb8h:
W1-Limited innovation: likely not addressed.
W2-Scope limited to node-level tasks: partially addressed.
W3-Tuning limitations of SSL methods: likely not addressed.

**Reviewer Scores:**

For Reviewer 4GfW, the initial rating is 4, and it is likely to stay at 4.

For Reviewer CiDd, the initial rating is 4, and it is likely to stay at 4.

For Reviewer fb8h, the initial rating is 4, and it is likely to stay at 4.

---

### Decision · Program_Chairs · 2026-01-26

Reject